# Two-Stage Feature Generator for Handwritten Digit Classification

**DOI:** 10.3390/s23208477

**Published:** 2023-10-15

**Authors:** M. Altinay Gunler Pirim, Hakan Tora, Kasim Oztoprak, İsmail Butun

**Affiliations:** 1Vakifbank, 06200 Ankara, Turkey; minealtinay.gunler@vakifbank.com.tr; 2Department of Avionics, Atilim University, 06830 Ankara, Turkey; hakan.tora@atilim.edu.tr; 3Department of Computer Engineering, Konya Food and Agriculture University, 42080 Konya, Turkey; 4Department of Computer Engineering, KTH Royal Institute of Technology, SE-114 28 Stockholm, Sweden; 5Department of Computer Engineering, OSTIM Technical University, 06370 Ankara, Turkey

**Keywords:** minimum distance classifier, neural network, principal component analysis, support vector machine, pattern recognition, soft sensor

## Abstract

In this paper, a novel feature generator framework is proposed for handwritten digit classification. The proposed framework includes a two-stage cascaded feature generator. The first stage is based on principal component analysis (PCA), which generates projected data on principal components as features. The second one is constructed by a partially trained neural network (PTNN), which uses projected data as inputs and generates hidden layer outputs as features. The features obtained from the PCA and PTNN-based feature generator are tested on the MNIST and USPS datasets designed for handwritten digit sets. Minimum distance classifier (MDC) and support vector machine (SVM) methods are exploited as classifiers for the obtained features in association with this framework. The performance evaluation results show that the proposed framework outperforms the state-of-the-art techniques and achieves accuracies of 99.9815% and 99.9863% on the MNIST and USPS datasets, respectively. The results also show that the proposed framework achieves almost perfect accuracies, even with significantly small training data sizes.

## 1. Introduction

Pattern recognition typically involves both feature generation and classification. In pattern recognition approaches, such as face recognition and digit recognition, a feature extractor aims to find the characteristics of patterns that can discriminate and separate classes. However, a variability of features can lead to difficulties in such approaches. For example, even if it is desirable to have small within-class variability in a face recognition approach, varying lighting conditions can lead to differences in features. Similarly, digits written by different people in digit recognition systems can cause variability in features [1,2,3,4,5,6,7]. Hence, determining and using the most efficient framework for feature generation and classification is crucial in the pattern recognition approaches. 

There are studies conducted on feature generation and classification in the literature. For instance, linear transformation techniques, such as principal component analysis (PCA), singular value decomposition (SVD), independent component analysis, discrete Fourier transform, Hadamard and Haar transforms, and discrete time wavelet transform (DTWT), are used for feature generation in the literature [7]. Moreover, neural networks (NNs) are used for classification in numerous studies such as in [2,3,4,5,6,7]. It should be stressed that typical recognition architectures use a single feature extractor followed by a supervised classifier. However, as stated in [8,9], two successive stages of feature generation yield higher accuracies than a one-stage extractor. There are also studies in the literature in which two or more feature extractors are cascaded, and the resulting features are used to train a supervised classifier. Even though such studies are exploited for the feature generation and classification in the literature, they cannot deal with within-class variability with a high performance which may cause difficulties for them to distinguish features into different classes.

In this paper, a new feature extraction method is presented. The method uses two consecutive attribute extractors. The first one generates the projected patterns on principal components or eigenvectors obtained from the covariance matrix of the data via PCA. The second provides the hidden layer outputs of a partially trained neural network (PTNN), where training of the neural network is stopped after a few epochs, i.e., the training is not fully completed. These two generators are cascaded, that is, the outputs of the PCA stage become the inputs of PTNN. We show that the proposed feature generator reduces within-cluster variability. This makes it much easier to distinguish data from different classes. The original input data are first transformed into a new space referred to as the PCA feature space. Then, the feature space is transformed into another space through a PTNN with one hidden layer with various hidden units. We show that a two-stage feature generator is advantageous in terms of the distribution of clusters in the feature space. 

It should be stressed here that the framework proposed in this study reduces within-cluster variability as compared to state-of-the-art studies. In this way, it becomes much easier to differentiate data from different classes using the framework. Moreover, the proposed framework can enable achieving low intra-class and high inter-class variations. In addition to all these advantages, more importantly, the proposed framework achieves the best performance on the MNIST and USPS handwritten digit datasets compared to all the studies in the literature. To assess the clusterability of the features generated using the proposed method, minimum distance classifier (MDC) and support vector machine (SVM) are used as classifiers.

This paper is organized as follows. Section 2 presents state-of-the-art studies in the literature. The proposed framework is discussed in Section 3. Section 4 discusses the verification of intra-class and inter-class feature distributions. The experimental results are described in Section 5. Finally, Section 6 concludes the study.

## 2. State of the Art

There are studies based on handwritten character recognition in the literature. For instance, Mellouli et al. [1] proposed a new convolutional neural network (CNN) architecture using morphological filters for digit recognition. The morphological configuration was called Morph-CNN, which achieved a test accuracy of 99.66% on the MNIST dataset. Patel et al. proposed a multi-resolution technique using a discrete wavelet transform (DWT)-based approach for handwritten character recognition [10]. The authors used the DWT to extract features and they also used the MDC to recognize the system output. Their technique achieved an overall success rate of 90.00%. Ayyaz et al. [11] proposed a hybrid feature extraction system based on the SVM. Their system was tested on both handwritten digits and uppercase alphabets, which achieved higher efficiency compared to other methods. Shubhangi et al. [12] proposed a structural micro-feature system based on the SVM to recognize handwritten English characters and digits with a high recognition rate. 

Liu et al. [13] proposed an NN-based system, which achieved improved accuracy by discriminative training and achieved a 98.45% recognition rate on the CENPARMI dataset. Suen et al. [14] developed a system to sort and identify cheques and financial documents on the CENPARMI dataset, which achieved a success rate of 98.85%. Lee et al. [15] proposed an offline handwritten digit recognition system for the CEDAR dataset, which achieved a recognition rate of 99.09%. Filatov et al. [16] designed a system based on an address script to identify handwritten postal addresses for US mail on the CEDAR dataset, which achieved a success rate of 99.54%. 

In [17], a discriminative cascaded CNN model was used, which achieved an error rate of 0.18% on the MNIST dataset. Ganapathy et al. [18] studied a multiscale NN recognition system. In [19], a single-layer NN achieved a 98.39% accuracy on the proposed MNIST dataset. In [20], four different techniques, i.e., the PCA, CNN, SVM, and multi-classifier systems, were used to develop a powerful system for handwritten character recognition, which achieved a success rate of 98.50% on the MNIST dataset. In [21], a cascaded PCA, binary hashing, and block-wise histograms were used with a very simple deep learning network for image classification, which achieved a 99.67% recognition rate. In [22], a system based on a multicolumn deep neural network (MCDNN) was developed using 35 pre-trained CNNs, which achieved an error rate of 0.23% on the MNIST dataset. Bruna et al. [9] used an invariant scattering convolution network, which achieved an error rate of 0.43% on the MNIST dataset. Goodfellow et al. [23] used a convolution max-out system to regularize dropout, which achieved an error rate of 0.45% on the MNIST dataset. Zeiler et al. [24] proposed stochastic pooling on deep CNN, which achieved an error rate of 0.47% on the MNIST dataset. In [25], a context-dependent deep NN/hidden Markov model was used for large-vocabulary speech recognition. This system was tested on both the MNIST and TIMIT datasets, which achieved an error rate of 0.83% on the MNIST dataset. Jarrett et al. [8] used large CNNs and achieved an error rate of 0.53% on the MNIST dataset without distortions. Yu et al. [26] used a hierarchical two-layer sparse coding network on pixels, which achieved an error rate of 0.77% on the MNIST dataset. Keysers et al. [27] proposed an image distortion model based on local optimization, which achieved a low error rate of 0.54% on the MNIST dataset. In [28], a scalable generative model based on a convolutional deep belief network was used for unlabeled data from the MNIST dataset, which achieved an error rate of 0.82%. 

In [29], pattern recognition using average patterns of categorical k-nearest neighbors was proposed, which achieved error rates of 1.27% and 3.44% on the MNIST and USPS datasets, respectively, using kernel classification on categorical average patterns. In [30], a discriminative-based supervised dictionary learning was developed, which achieved test error rates of 0.60% and 2.40% for the MNIST and USPS datasets, respectively. Error rates of 1.66% and 2.59% were achieved using SVM and KNN, respectively, on the USPS test set in [31]. In [32], perceptron learning of a modified quadratic discriminant function (MQDF) was used to achieve error rates of 1.49% and 2.19% on the MNIST and USPS datasets, respectively, which indicates that discriminative learning of MQDF can further improve MQDF’s performance. Xu et al. [33] presented a nonnegative representation-based classifier for pattern classification, which achieved accuracies of 99% and 95.1% on the MNIST and USPS datasets, respectively. Prasad et al. [34] presented novel features and cascaded classifiers KNN and SVM, resulting in an accuracy of 99.26 on the MNIST dataset.

## 3. Proposed Framework

Employing appropriate features to classify data can directly influence desired learning results. Therefore, selecting and generating features that are easily separable is vital for accurate classification [1,2,3]. Considering this motivation, a two-stage cascaded feature generator framework is proposed in this study. 

In this sub-section, first, a one-stage feature generator, which provides the basis for the proposed framework, is discussed, and then the proposed two-stage feature generator framework of this study is introduced.

### 3.1. Soft Sensor Implementation for the Feature Generation

The proposed method has been implemented by using both hardware sensors (cameras, scanners, etc.) and soft sensors. The former captures the digits. The latter provides features that no hardware sensor is able to measure. In this study, a soft sensor model was developed to generate features for handwritten digit classification. The soft sensor has been realized by two cascaded modules, namely PCA and PTNN. The following section presents the details of each module. 

Handwritten digit classification (HDC) has found such applications as postal automation, bank check automation, and human–computer interactions in practice. Many studies have been conducted for the classification of digits, as mentioned in Section 2. The first and most vital step in the recognition cycle is the collection of handwritten digits from people. There exist various ways to acquire the digits depending on the way the digits are generated. Therefore, different sensors are utilized for capturing the digits. While the digits written on paper can be recorded by handheld scanners or cameras, the digits created in the air can be captured by Kinect cameras, wearable inertial measurement unit (IMU) sensors, and wearable smart gloves and armbands. They rely on capturing hand and finger movements. In addition, a smart pen that exploits the inertial force sensors can record the digits [35,36,37,38].

### 3.2. One-Stage Feature Generator

Figure 1 depicts a one-stage feature generator framework that employs the PCA for the feature extraction. As can also be seen from the figure, the implementation of the one-stage classifier is based on either the MDC, which is a simple algorithm, or the SVM, which is a sophisticated algorithm, for classification [11,12]. MDC can be defined as calculating the distance between the unknown data and each class center and assigning the data to the nearest class center with the shortest Euclidean distance.

Algorithm 1, which is presented below, describes the steps to generate the features based on the PCA within this framework.
**Algorithm 1:** Obtaining principal component (PC)-based features$\mathrm{The}\;\mathrm{data}\;\mathrm{matrix}\;D=\left(d_{1}\;d_{2}\ldots d_{N}\right)$ of size MxN$\;\mathrm{where}\;d_{i}$$\;\mathrm{represents}\;\mathrm{the}\;i^{th}$ sample from the data matrix, where i = 1, …, N and N is the number of examples in data matrix.S1: Scale the values of the data matrix in [0, 1]$.\;\mathrm{Resulting}\;\mathrm{matrix}\;\mathrm{is}\;\mathrm{called}\;D_{s}$.$S2:\;\mathrm{Calculate}\;\mathrm{the}\;\mathrm{principal}\;\mathrm{components}\;(\mathrm{PCs})\;\mathrm{of}\;\mathrm{the}\;D_{s}$.S3: Select the PCs corresponding to K highest eigenvalues.$S4:\;\mathrm{Construct}\;\mathrm{the}\;\mathrm{matrix}\;\mathrm{whose}\;\mathrm{columns}\;\mathrm{are}\;\mathrm{formed}\;\mathrm{by}\;\mathrm{the}\;\mathrm{principal}\;\mathrm{component}\;\mathrm{vectors}\;(\mathrm{eigenvectors}\;\mathrm{of}\;D_{s}$$)\;C\;=\;\left(c_{1}\;c_{2}\;\ldots c_{K}\right)$ with the size of MxK.S5: Calculate the feature matrix by      $F\;=\;D_{s}^{T}\;C\;=\;\left(d_{1}c_{1}\;d_{1}c_{2}\;\ldots d_{1}c_{K}\right)$ with the size of NxK. Although the algorithm ends in this step, the following step demonstrates the effectiveness of the generated features.S6: Train a classifier (such as SVM or MDC) by the rows of the matrix F.


It is easy to show that the elements of matrix ***F*** are the projection of each data sample on principal component vectors. In ***F***, the product $d_{i}c_{j}$ denotes the inner product of the two vectors. Hence, we can express this product as:(1)$$\langle d_{i},c_{j}\rangle =∥d_{i}∥∥c_{j}∥\cos\theta$$
where θ is the angle between $d_{i}$ and $c_{j}$, i=1, …, N and j=1, …, K.

Since $∥c_{j}∥=1$, the inner product can be written as:(2)$$\langle d_{i},c_{j}\rangle =∥d_{i}∥\cos\theta$$

Equation (2) represents the projection of $d_{i}$ and $c_{j}$, that is:(3)$$proj_{c_{j}}d_{i}=∥d_{i}∥\cos\theta$$

Consequently, the projected data are employed as features to train the selected classifier, which is based on the MDC or SVM.

It is a fact that the variance in clusters obtained from using the PCA is very large; hence, the MDC or SVM classifiers cannot successfully separate one cluster from another. This is due to features sparsely scattered around the center of the cluster (i.e., distances between samples within the same cluster are high). This reduces the classification performance which yields low success rates.

### 3.3. Two-Stage Feature Generator

To enhance the performance of the one-stage generator, we propose inserting another transformation operator between the PCA and MDC/SVM modules to form a two-stage feature generator framework. Figure 2 depicts the proposed framework. The framework for a two-stage feature generator is explained in more detail in Algorithm 2 step by step.

The PTNN module in the framework is simply a multilayer perceptron (MLP) [2] with one hidden layer with various neurons. It is structured for the purpose of classification. Thus, the outputs of the network correspond to the clusters to be identified, i.e., the number of digits in our test cases. The network is fed by the projected data features. However, the network is not fully trained, but partially trained. Therefore, training is halted after a few epochs. The epoch errors are high, which indicates that training is far from complete. When training is halted, the network cannot correctly identify the clusters. However, we keep on training the network to observe the behavior of the PTNN at the early stage of the training. In summary, PTNN is simply an MLP without full training, or the training period is stopped after a predefined number of epochs.

Figure 3a,b illustrate mean squared error (MSE) results obtained from the fully trained NN and PTNN training, respectively. MSE is computed as the mean of the squared differences between the actual output and the estimated output. Figure 3b represents the performance of the neural network at the early stages of training. The results show that the MSE decreases rapidly at the beginning of the training phase and changes slowly until 2000 epochs are reached. Then, it remains almost constant, implying that the NN is fully trained. In total, 60,000 and 10,000 samples are employed during the training and testing phases, respectively, and a test accuracy of 98.58% is achieved [39]. Additionally, when an MLP NN is trained for classifying the digits in the MNIST dataset with zero feature extraction, the number of the epochs required varies from 40 to 50 to achieve test accuracies between 87% and 98% using 60,000 samples for the training set and 10,000 samples for the test set [40,41].

Despite stopping the training at a significantly early stage, if the outputs of the hidden unit of the partially trained network are used as features, we find that intra-cluster distances are reduced as compared to those in the PCA feature space. On the other hand, the size of the feature vectors in the two-stage feature generator is higher than those in the one-stage feature generator (i.e., larger than K). That is, the feature space composed of the two-stage feature generator includes more features than that of the one-stage feature generator. Hence, the proposed approach does not reduce the number of features. However, it improves the accuracy of the classifier. 

Algorithm 2 describes the transformation to generate features based on the PCA plus PTNN.
**Algorithm 2:** Obtaining neural network-based features from projected data on the PCs$\mathrm{The}\;\mathrm{feature}\;\mathrm{matrix}\;F\;=\;\left(f_{1}\;f_{2}\;\ldots \;f_{K}\right)$S1: Build an MLP network with one hidden layer and P hidden nodes (neuron).S2: Start training the network for classifying the examples represented by the rows of F.S3: Halt training in early iterations.S4: Calculate the outputs of the hidden layer$$h_{i}\;=\;sigmoid\left(FW+b\right)\;$$where, W is the weight matrix between input and hidden layers and i = 1, …, P$S5:\;\mathrm{Construct}\;\mathrm{the}\;\mathrm{hidden}\;\mathrm{layer}\;\mathrm{output}\;\mathrm{matrix}\;H\;=\;\left(h_{1}\;h_{2}\;\ldots \;h_{P}\right)$ whose size is NxP. Although the algorithm ends in this step, the following step demonstrates the effectiveness of the generated features. S6: Train a classifier (such as SVM or MDC) by the rows of the matrix H.

The algorithms discussed above are tested on the MNIST and USPS digit datasets to analyze the distance distribution of each digit class in this study. For this purpose, the distances within the class and between classes are calculated, where the Euclidean distance is used as the distance metric. Let $d_{i}$ and $d_{j}$ be row vectors in $R^{N}$. Then, the Euclidean distance between these two vectors is defined as
(4)$$\;L=\left|\left|d_{j}-d_{i}\right|\right|=sqrt\left[\;{\left(d_{j1}-d_{i1}\right)}^{2}+\ldots +{\left(d_{jN}-d_{iN}\right)}^{2}\;\right]$$

The within-cluster distances are calculated by Algorithm 3.
**Algorithm 3:** Calculating the distances among the feature vectors within a digit class$\mathrm{Assume}\;\mathrm{that}\;F^{m}\;=\;\left(f_{1}\;f_{2}\;\ldots f_{S}\right)$$\;\mathrm{is}\;a\;\mathrm{feature}\;\mathrm{matrix}\;\mathrm{for}\;m^{th}$ class and m = 1,2, …,O and S is the number of the examples in a given class.S1: Calculate the centroid of the class m as:$F_{c}^{m}=\frac{1}{S}\overset{S}{\underset{s=1}{\sum }}f_{s}$S2: Calculate the Euclidean distance between each example and centroid vectors as:$L_{mc}=\left|\left|f_{m}-F_{c}^{m}\right|\right|$


It is envisaged that the proposed framework should yield minimized intra-cluster distances or maximized inter-class distances. This envisagement is proved in the following section by considering both the one-stage and two-stage feature generators and the algorithms considered.

## 4. Verification of Inter and Intra Class Distributions

In this section, the intra-class and inter-class distance distributions are verified using the distance metric presented in Equation (5). To form the metric, first of all, the standard deviation, which indicates how sparsely or densely distributed the distances are within a class, is determined for each class. Then, to quantify the distance between the classes, the separation metric (SM) is formed:(5)$$SM=\frac{d_{ij}}{\left(\sigma_{i}+\sigma_{j}\right)/2}$$
where $d_{ij}$ is the distance between the centers of classes *i* and *j*, while $\sigma_{i}$ and $\sigma_{j}$ are the standard deviations for classes *i* and *j*, respectively. This metric represents the degree of separability. The inter-class distances are calculated by Algorithm 4.
**Algorithm 4:** Calculating the distances between the two-digit classes$\mathrm{Assume}\;\mathrm{that}\;F^{m}\;=\;\left(f_{1}\;f_{2}\;\ldots f_{S}\right)\;$$\mathrm{is}\;a\;\mathrm{feature}\;\mathrm{matrix}\;\mathrm{for}\;m^{th}$class and m = 1,2, …, O and S is the number of the examples in a given class.S1: Calculate the centroids for each class in a given dataset.S2: Calculate the distance between the centroids of two classes.$L_{m\left(m-1\right)}=\left|\left|F_{c}^{m}-F_{c}^{m-1}\right|\right|$


In step 2 of Algorithm 4, m(m−1) represents the distance of the center of each class from the centers of all other classes. Suppose the following:Case 1: if the distance remains constant and the standard deviations in Equation (5) have small values, then SM becomes higher. Note that a higher SM indicates better separation.Case 2: if the standard deviations in Equation (5) are constant and the distance has high values, then SM becomes higher.

These two cases are illustrated in Figure 4. Table 1 and Table 2 show the standard deviations of distances between the centers of classes using the one-stage and two-stage feature generators, respectively, in the USPS dataset. The results show that the standard deviations (i.e., σ’s) using the two-stage feature generator are smaller than those using the one-stage feature generator. This is associated with the fact that samples in the given class are distributed close to the center of the class. A consequence of this is that the data are more separable in the feature space formed by the PCA plus PTNN. In other words, the boundary or volume of each cluster shrinks inward. On the other hand, with the one-stage case, the samples in each class are scattered away from the center of the class so that small values of standard deviations are obtained. Consequently, the variation within a cluster without the PTNN is higher than that with the PTNN.

Table 3 and Table 4 show the separability values calculated by Equation (5) for the one-stage and two-stage feature generators, respectively. It can be seen that classes scattered in the feature space are more separable in the two-stage case (i.e., the separability increases). In pattern recognition, this is one of the desired requirements for a classifier to classify data accurately. Furthermore, we can obtain the *SM* ratio of the value of a selected class from Table 4 to the value of that class in Table 3. Once these ratios are calculated, it can be seen from Table 5 that they are mostly greater than 1. Thus, the classes in the feature space built from the PCA plus NN are more separable compared to those in the PCA space.

The same cluster behavior is also observed for the MNIST digit dataset. Table 6 and Table 7 show the standard deviations for the clusters formed with 5000 and 10,000 samples, respectively. It can be seen from the tables that the variation within a cluster with the two-stage extractor is lower than that with the one-stage generator. The separability values and SM ratios for the MNIST dataset are shown in Table 8, Table 9 and Table 10, respectively.

## 5. Results and Discussion

The performance of the proposed feature generator is tested on the MNIST and USPS digit datasets. The USPS handwritten digit dataset is derived from a project on recognizing handwritten digits on envelopes [42]. The digits have sizes of 16 × 16 pixels. It contains 7291 samples for the training set and 2007 samples for the test set. The standard MNIST dataset is derived from the NIST dataset and was created by LeCun et al. [43]. The digits have sizes of 28 × 28 pixels. It has 60,000 samples for the training set and 10,000 samples for the test set. Figure 5 and Figure 6 show some examples of the digits from the MNIST and USPS datasets, respectively. The MDC and SVM are utilized to identify digits in these datasets. The MDC is a simple classifier. In the training phase, training vectors are separated by each class. Then, the mean values of each class are computed. In the test phase, the closest mean to the test vector is calculated via the Euclidean distance. Then, the corresponding class is predicted. The SVM is much more complex than the MDC. It is capable of extracting not only linear but also curved decision boundaries. Thus, more accurate classification can be achieved by setting a maximum margin separator among the sample points, where the margin is defined as the distance of the decision boundary to the closest sample.

In Table 11, the results of the MDC for the USPS digit classes are shown for both the one-stage and two-stage feature generators. We then determine the accuracies for different eigenvalues and different training sizes. The results show that the best recognition rate is achieved using 4000 samples for the training set and 5298 samples for the test set with K = 10. Note that the NN is partly trained for various epochs, i.e., the training is halted in the early stage of iterations. As an example, Table 12 presents the accuracies for K=10 at different epochs and different training sizes. The table shows that the performance of PCA plus PTNN (two-stage generator) is higher than that of the one-stage extractor. Moreover, as an example, the performance of the two-stage generator framework with a training size of 500 samples is improved by 2.386 points with reference to the one-stage extractor at an epoch of 15 for the USPS dataset. During the training for each scenario, the learning rate and the number of hidden nodes are set to 0.5 and 50, respectively. Then, hidden layer outputs are extracted from the NN. The mean values of these outputs are calculated for each digit class. For the unseen test data, the hidden layer output is calculated. Then, Euclidean distances of the test data to the mean values of digit classes are computed. The test data are classified according to the digit class with minimum distance. For all the scenarios, two-stage features lead to higher performance than one-stage features. The average test recognition rates for 10 classes are 91.60% and 90.13% at the training size of 4000 for the two-stage and one-stage cases, respectively.

Table 13 presents the performance rates for the MNIST digit classes. As seen, the performance of a two-stage extractor is lower than that of the one-stage extractor for small training sizes. However, an improvement in the performance appears for the full training size of 60,000.

Table 14 and Table 15 show the test success rates of the SVM classifier for the USPS and MNIST datasets, respectively. The experiments on SVM are held with the RBF kernel function. Although the best performance is obtained using 60,000 samples for the training set and 10,000 samples for the test set, it is clear that small training sizes also result in very high accuracies. PTNN is trained with a learning rate of 0.50 and 50 hidden nodes.

Table 16 lists the accuracies with K=8 at different epochs and training sizes. The improvements in the performance of the two-stage extractor are clear; for instance, accuracy is increased by 1.5869 points with respect to the one-stage extractor at an epoch of 10 for a training size of 5000.

Although the PTNN is trained for 5% to 30% of the MNIST and USPS datasets, the proposed method achieves almost perfect performance with the SVM. Furthermore, the performance is acceptable even for a simple MDC. The results show that the proposed approach provides more relevant features for the data. Hence, the classifier achieves much better performance scores, i.e., 99.9863% and 99.9815%, for the USPS and MNIST datasets, respectively. To the best of our knowledge, these are the best performances in the current literature. 

Table 17 shows the effectiveness of the two-stage feature extractor. Improvements in the accuracies with respect to the one-stage extractor are clear for each classifier. This shows that the proposed features give quite better abilities of generalization to the classifiers.

Table 18 and Table 19 show comparisons of the performances of our framework and some state-of-the-art methods on the MNIST and USPS datasets, respectively. The results show that the proposed method outperforms well-known techniques in the literature. Note that the SVM using two-stage features achieves error rates of 0.0185% and 0.0137% for the MNIST and USPS datasets, respectively, which are currently the best performances in the literature. 

## 6. Conclusions and Future Work

In this paper, we proposed a novel framework based on a two-stage feature generator for handwritten digit classification. The first stage of this framework relies on the PCA, which generates the projected data from the eigenvectors corresponding to the highest K eigenvectors. The second stage has been constructed by a PTNN whose training has been halted at early epochs, i.e., it was not fully trained to recognize the input classes. This PTNN has been fed by the projected data on principal components and then its hidden layer outputs have been selected as new features, which have been used to train the MDC and SVM classifiers. 

We evaluated the performance of the proposed method on the MNIST and USPS datasets. In both datasets, the best results are performed by using an SVM classifier. We found out that the two-stage feature extractor has led to noticeable improvements in terms of accuracy. Moreover, compared to current state-of-the-art methods, the proposed framework has resulted in almost perfect performances even with small training sizes. In addition, our experiments have shown that the proposed method can achieve error rates of 0.0185% and 0.0137% for the MNIST and USPS datasets, respectively, which can currently be considered the best performances in the literature.

In future work, as an easier but meaningful expansion, sign recognition will be added to the study. As a more complex study, we will use face and texture datasets to further evaluate the usefulness of our proposed framework.

## Figures and Tables

**Figure 1 sensors-23-08477-f001:**
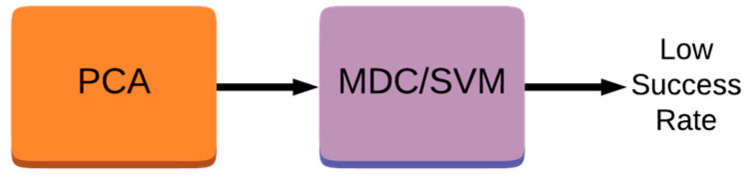
Structure of one-stage feature generator with handwritten digit inputs.

**Figure 2 sensors-23-08477-f002:**
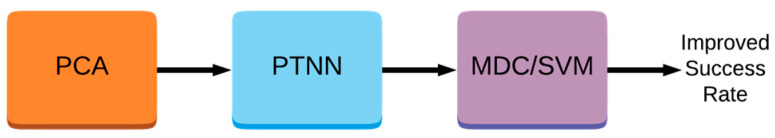
Structure of two-stage feature generator with handwritten digit inputs.

**Figure 3 sensors-23-08477-f003:**
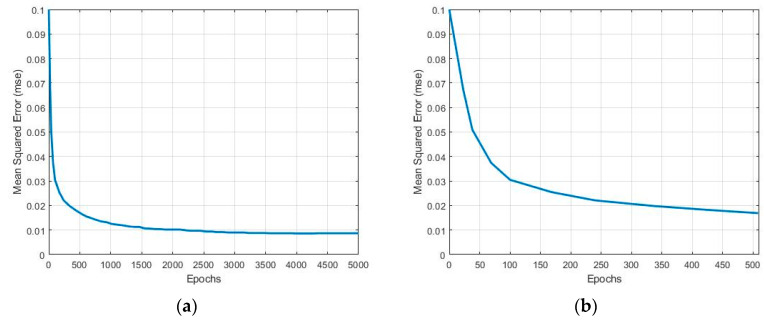
MSE of fully (**a**) and partially (**b**) trained neural network.

**Figure 4 sensors-23-08477-f004:**
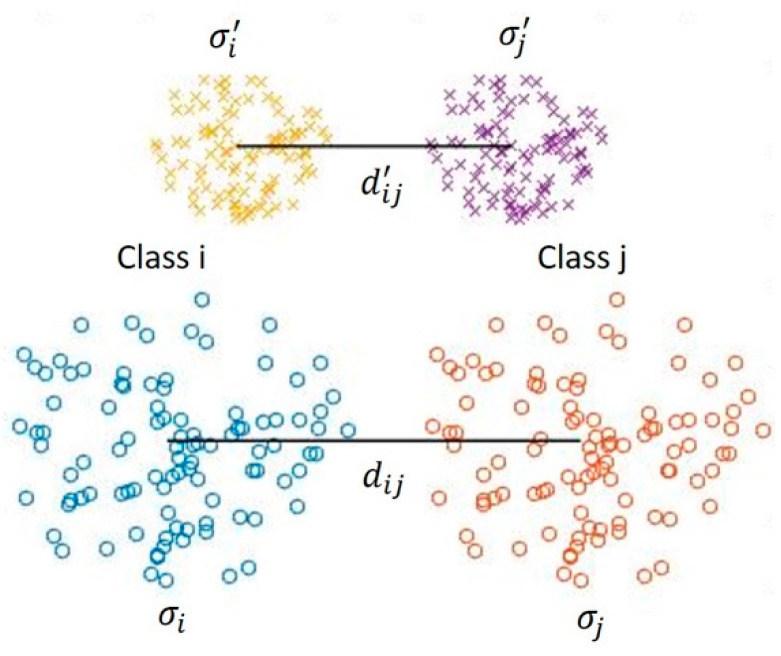
Representative distribution of features in space with high and low standard deviations.

**Figure 5 sensors-23-08477-f005:**
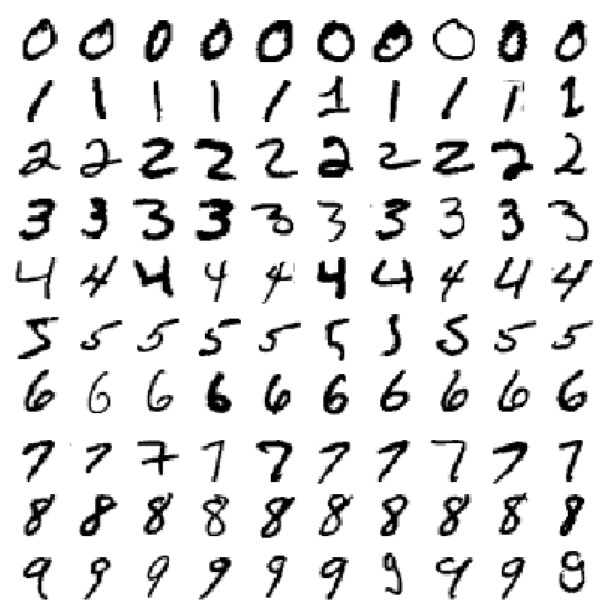
Samples of digits from the MNIST dataset.

**Figure 6 sensors-23-08477-f006:**
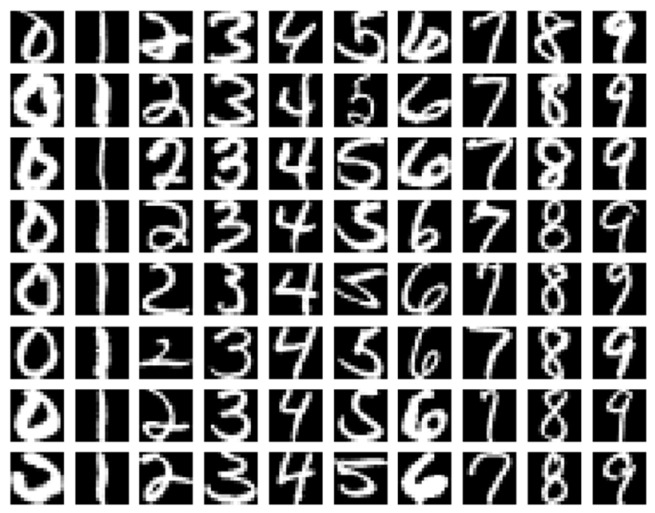
Samples of digits from the USPS dataset.

**Table 1 sensors-23-08477-t001:** Standard deviations for each digit class in USPS with a training size of 500.

Stage/Cluster	Digit 0	Digit 1	Digit 2	Digit 3	Digit 4	Digit 5	Digit 6	Digit 7	Digit 8	Digit 9
One stage	0.7507	0.5342	0.6096	0.6058	0.8084	0.7281	0.7214	0.8974	0.5227	0.8824
Two stage (50 epoch)	0.3003	0.3133	0.4270	0.2696	0.3627	0.4006	0.3733	0.5240	0.3243	0.3276
Two stage (30 epoch)	0.3272	0.3248	0.3920	0.3023	0.3238	0.4337	0.3691	0.5604	0.2510	0.2918
Two stage (25 epoch)	0.3439	0.2900	0.3940	0.2795	0.3075	0.3657	0.3254	0.5133	0.3002	0.3253
Two stage (20 epoch)	0.3814	0.3038	0.4293	0.3132	0.3055	0.3650	0.3218	0.4948	0.2516	0.3534
Two stage (15 epoch)	0.3727	0.2299	0.4077	0.3572	0.3299	0.4197	0.3126	0.4213	0.3078	0.3053
Two stage (10 epoch)	0.3319	0.2372	0.3516	0.3315	0.3709	0.3883	0.3709	0.4483	0.2579	0.3074

**Table 2 sensors-23-08477-t002:** Standard deviations for each digit class in USPS with a training size of 1000.

Stage/Cluster	Digit 0	Digit 1	Digit 2	Digit 3	Digit 4	Digit 5	Digit 6	Digit 7	Digit 8	Digit 9
One stage	0.7689	0.5903	0.5389	0.6551	0.7750	0.7185	0.7386	0.9349	0.6892	0.8449
Two stage (50 epoch)	0.3509	0.3345	0.3659	0.3366	0.3916	0.4667	0.3281	0.5737	0.3655	0.2778
Two stage (30 epoch)	0.3194	0.3415	0.3356	0.3410	0.3763	0.3280	0.3356	0.4825	0.3035	0.2602
Two stage (25 epoch)	0.3912	0.3582	0.3861	0.3695	0.3686	0.4229	0.3287	0.5040	0.3699	0.3368
Two stage (20 epoch)	0.3706	0.3396	0.3503	0.3869	0.3784	0.4396	0.3620	0.5676	0.3500	0.3260
Two stage (15 epoch)	0.3186	0.3245	0.3887	0.3901	0.4236	0.4273	0.3295	0.4660	0.2948	0.2926
Two stage (10 epoch)	0.4081	0.3260	0.3418	0.3785	0.3152	0.3587	0.3313	0.4718	0.3157	0.3764

**Table 3 sensors-23-08477-t003:** Separability values for only PCA with a training size of 500.

Digits	Digit 0	Digit 1	Digit 2	Digit 3	Digit 4	Digit 5	Digit 6	Digit 7	Digit 8	Digit 9
Digit 0	0	9.8748	5.8467	5.8775	5.9938	4.7561	5.3691	6.3096	6.8799	6.0279
Digit 1	9.8748	0	6.8908	8.2924	6.4615	7.1265	6.5954	5.9076	7.8357	5.5381
Digit 2	5.8467	6.8908	0	5.3733	4.8058	4.6786	4.2704	4.9557	5.0231	4.8123
Digit 3	5.8775	8.2924	5.3733	0	5.4038	3.7772	5.6887	4.9538	5.3137	4.5252
Digit 4	5.9938	6.4615	4.8058	5.4038	0	4.3211	4.7730	3.9199	4.7242	2.9392
Digit 5	4.7561	7.1265	4.6786	3.7772	4.3211	0	3.8773	4.5386	4.4424	3.9667
Digit 6	5.3691	6.5954	4.2704	5.6887	4.7730	3.8773	0	5.6352	5.5718	5.3602
Digit 7	6.3096	5.9076	4.9557	4.9538	3.9199	4.5386	5.6352	0	4.9600	2.3206
Digit 8	6.8799	7.8357	5.0231	5.3137	4.7242	4.4424	5.5718	4.9600	0	3.7385
Digit 9	6.0279	5.5381	4.8123	4.5252	2.9392	3.9667	5.3602	2.3206	3.7385	0

**Table 4 sensors-23-08477-t004:** Separability values for only PCA plus NN with a training size of 500.

Digits	Digit 0	Digit 1	Digit 2	Digit 3	Digit 4	Digit 5	Digit 6	Digit 7	Digit 8	Digit 9
Digit 0	0	11.3646	6.3528	9.2070	8.6145	6.5908	7.0915	8.1459	8.1696	9.9020
Digit 1	11.3646	0	7.7678	10.2732	8.4149	8.0777	9.9724	7.5722	8.6099	8.6258
Digit 2	6.3528	7.7678	0	7.0777	7.1212	6.2013	6.1736	5.9118	5.7625	7.6732
Digit 3	9.2070	10.2732	7.0777	0	8.4951	6.3814	9.6334	6.6971	6.9023	7.8166
Digit 4	8.6145	8.4149	7.1212	8.4951	0	6.2550	8.3657	6.1707	6.5198	5.7475
Digit 5	6.5908	8.0777	6.2013	6.3814	6.2550	0	5.9684	6.2361	5.5147	6.9847
Digit 6	7.0915	9.9724	6.1736	9.6334	8.3657	5.9684	0	8.5955	8.0326	10.4945
Digit 7	8.1459	7.5722	5.9118	6.6971	6.1707	6.2361	8.5955	0	5.9426	4.4272
Digit 8	8.1696	8.6099	5.7625	6.9023	6.5198	5.5147	8.0326	5.9426	0	6.5642
Digit 9	9.9020	8.6258	7.6732	7.8166	5.7475	6.9847	10.4945	4.4272	6.5642	0

**Table 5 sensors-23-08477-t005:** Separability ratios of PCA + NN to PCA with a training size of 500.

Digits	Digit 0	Digit 1	Digit 2	Digit 3	Digit 4	Digit 5	Digit 6	Digit 7	Digit 8	Digit 9
Digit 0	0	1.1509	1.0866	1.5665	1.4372	1.3858	1.3208	1.2910	1.1875	1.6427
Digit 1	1.1509	0	1.1273	1.2389	1.3023	1.1335	1.5120	1.2818	1.0988	1.5575
Digit 2	1.0866	1.1273	0	1.3172	1.4818	1.3255	1.4457	1.1929	1.1472	1.5945
Digit 3	1.5665	1.2389	1.3172	0	1.5721	1.6894	1.6934	1.3519	1.2990	1.7273
Digit 4	1.4372	1.3023	1.4818	1.5721	0	1.4475	1.7527	1.5742	1.3801	1.9555
Digit 5	1.3858	1.1335	1.3255	1.6894	1.4475	0	1.5393	1.3740	1.2414	1.7608
Digit 6	1.3208	1.5120	1.4457	1.6934	1.7527	1.5393	0	1.5253	1.4416	1.9579
Digit 7	1.2910	1.2818	1.1929	1.3519	1.5742	1.3740	1.5253	0	1.1981	1.9078
Digit 8	1.1875	1.0988	1.1472	1.2990	1.3801	1.2414	1.4416	1.1981	0	1.7558
Digit 9	1.6427	1.5575	1.5945	1.7273	1.9555	1.7608	1.9579	1.9078	1.7558	0

**Table 6 sensors-23-08477-t006:** Standard deviations for each digit class in MNIST with a training size of 5000.

Stage/Cluster	Digit 0	Digit 1	Digit 2	Digit 3	Digit 4	Digit 5	Digit 6	Digit 7	Digit 8	Digit 9
One stage	0.8528	0.8807	0.7602	0.8453	0.8398	0.8349	0.9544	0.9431	0.8829	0.9686
Two stage (50 epoch)	0.4402	0.3983	0.4066	0.4055	0.3644	0.3150	0.3390	0.4530	0.3436	0.4012
Two stage (30 epoch)	0.4377	0.4302	0.3775	0.4428	0.3153	0.3705	0.3999	0.5035	0.3684	0.3494
Two stage (25 epoch)	0.4074	0.4057	0.3848	0.4336	0.3475	0.3611	0.4249	0.4539	0.3450	0.4115
Two stage (20 epoch)	0.4173	0.3745	0.3910	0.3960	0.3703	0.2929	0.3864	0.4346	0.3773	0.4278
Two stage (15 epoch)	0.4234	0.3822	0.3867	0.3810	0.3570	0.3651	0.4407	0.3893	0.3582	0.4257
Two stage (10 epoch)	0.3593	0.4270	0.3813	0.4151	0.3298	0.2729	0.4335	0.4621	0.3501	0.4343

**Table 7 sensors-23-08477-t007:** Standard deviations for each digit class in MNIST with a training size of 1000.

Stage/Cluster	Digit 0	Digit 1	Digit 2	Digit 3	Digit 4	Digit 5	Digit 6	Digit 7	Digit 8	Digit 9
One stage	0.8216	0.9208	0.7924	0.8668	0.8062	0.8346	0.9822	0.9548	0.8864	0.9897
Two stage (50 epoch)	0.3576	0.4009	0.3868	0.4353	0.3202	0.3851	0.3788	0.5001	0.3651	0.4702
Two stage (30 epoch)	0.4371	0.4241	0.4348	0.4427	0.3489	0.3826	0.4151	0.4505	0.3748	0.4831
Two stage (25 epoch)	0.3796	0.3756	0.4182	0.3315	0.4091	0.2733	0.4364	0.4433	0.3914	0.4414
Two stage (20 epoch)	0.4154	0.3689	0.4422	0.4776	0.4668	0.3932	0.3988	0.4533	0.4243	0.4175
Two stage (15 epoch)	0.3735	0.3422	0.4489	0.4090	0.3930	0.4430	0.3760	0.4891	0.3823	0.4644
Two stage (10 epoch)	0.8216	0.9208	0.7924	0.8668	0.8062	0.8346	0.9822	0.9548	0.8864	0.9897

**Table 8 sensors-23-08477-t008:** Separability values for only PCA with a training size of 5000.

Digits	Digit 0	Digit 1	Digit 2	Digit 3	Digit 4	Digit 5	Digit 6	Digit 7	Digit 8	Digit 9
Digit 0	0	8.9490	7.1822	6.7094	7.5911	5.4839	6.6199	7.1983	6.6764	6.9278
Digit 1	8.9490	0	6.1465	5.9734	6.7315	5.6308	6.2385	5.9189	5.3469	5.7358
Digit 2	7.1822	6.1465	0	5.5477	5.8153	5.6525	4.6556	6.3348	4.6635	5.6461
Digit 3	6.7094	5.9734	5.5477	0	6.4290	3.8336	6.2177	5.9367	4.1482	5.3524
Digit 4	7.5911	6.7315	5.8153	6.4290	0	5.0809	4.9329	4.5674	5.2593	2.9364
Digit 5	5.4839	5.6308	5.6525	3.8336	5.0809	0	4.8664	5.0609	3.7842	4.2076
Digit 6	6.6199	6.2385	4.6556	6.2177	4.9329	4.8664	0	5.9810	5.1489	4.8942
Digit 7	7.1983	5.9189	6.3348	5.9367	4.5674	5.0609	5.9810	0	5.3735	3.0897
Digit 8	6.6764	5.3469	4.6635	4.1482	5.2593	3.7842	5.1489	5.3735	0	4.1881
Digit 9	6.9278	5.7358	5.6461	5.3524	2.9364	4.2076	4.8942	3.0897	4.1881	0

**Table 9 sensors-23-08477-t009:** Separability values for only PCA + NN with a training size of 5000.

Digits	Digit 0	Digit 1	Digit 2	Digit 3	Digit 4	Digit 5	Digit 6	Digit 7	Digit 8	Digit 9
Digit 0	0	8.4546	6.3110	6.2432	8.1580	5.4891	7.8137	7.2026	6.2925	7.3703
Digit 1	8.4546	0	6.2979	6.1388	8.9035	7.8212	8.1683	6.8096	6.5099	7.5594
Digit 2	6.3110	6.2979	0	5.4763	7.1528	6.9759	6.4080	6.7837	5.8246	6.8386
Digit 3	6.2432	6.1388	5.4763	0	8.0550	5.3969	8.2691	6.1786	5.2527	6.5728
Digit 4	8.1580	8.9035	7.1528	8.0550	0	7.7562	6.9311	6.2156	7.0799	4.2345
Digit 5	5.4891	7.8212	6.9759	5.3969	7.7562	0	7.7062	7.2702	5.3668	6.9289
Digit 6	7.8137	8.1683	6.4080	8.2691	6.9311	7.7062	0	8.3848	7.5591	7.5293
Digit 7	7.2026	6.8096	6.7837	6.1786	6.2156	7.2702	8.3848	0	6.8602	4.2942
Digit 8	6.2925	6.5099	5.8246	5.2527	7.0799	5.3668	7.5591	6.8602	0	6.1857
Digit 9	7.3703	7.5594	6.8386	6.5728	4.2345	6.9289	7.5293	4.2942	6.1857	0

**Table 10 sensors-23-08477-t010:** Separability ratios of PCA + NN to PCA with a training size of 5000.

Digits	Digit 0	Digit 1	Digit 2	Digit 3	Digit 4	Digit 5	Digit 6	Digit 7	Digit 8	Digit 9
Digit 0	0	0.9448	0.8787	0.9305	1.0747	1.0009	1.1803	1.0006	0.9425	1.0639
Digit 1	0.9448	0	1.0246	1.0277	1.3227	1.3890	1.3093	1.1505	1.2175	1.3179
Digit 2	0.8787	1.0246	0	0.9871	1.2300	1.2341	1.3764	1.0709	1.2490	1.2112
Digit 3	0.9305	1.0277	0.9871	0	1.2529	1.4078	1.3299	1.0407	1.2662	1.2280
Digit 4	1.0747	1.3227	1.2300	1.2529	0	1.5265	1.4051	1.3609	1.3462	1.4421
Digit 5	1.0009	1.3890	1.2341	1.4078	1.5265	0	1.5836	1.4365	1.4182	1.6468
Digit 6	1.1803	1.3093	1.3764	1.3299	1.4051	1.5836	0	1.4019	1.4681	1.5384
Digit 7	1.0006	1.1505	1.0709	1.0407	1.3609	1.4365	1.4019	0	1.2767	1.3898
Digit 8	0.9425	1.2175	1.2490	1.2662	1.3462	1.4182	1.4681	1.2767	0	1.4770
Digit 9	1.0639	1.3179	1.2112	1.2280	1.4421	1.6468	1.5384	1.3898	1.4770	0

**Table 11 sensors-23-08477-t011:** Test recognition rates of MDC for USPS.

Training Size	500	1000	2000	4000	7291
Two-Stage	88.29	90.17	90.68	91.60	90.8451
One-Stage	84.89	86.89	88.80	90.13	89.2631

**Table 12 sensors-23-08477-t012:** Accuracies with K = 10 and various sizes and epochs for USPS.

Epoch/Training Size	500	1000	2000	4000
10	87.4890	89.4195	90.6508	91.3091
15	87.2824	89.4919	90.2459	91.5539
20	87.7258	89.7536	90.4792	91.4046
25	88.2981	89.3516	90.6851	91.5269
30	87.6763	89.3163	90.5018	91.5633
50	88.1997	90.1713	90.3087	91.6021
One-stage	84.8964	86.8968	88.8011	90.1319

**Table 13 sensors-23-08477-t013:** Test recognition rates of MDC for MNIST.

Training Size	5000	10,000	60,000
Two-Stage	93.4712	94.1145	97.2372
One-Stage	94.2240	95.3311	97.1316

**Table 14 sensors-23-08477-t014:** Test recognition rates of SVM for USPS.

Training Size	500	1000	2000	4000	7291
Two-stage	99.3362	99.72	99.79	99.922	99.9863
One-stage	98.26	98.08	98.42	97.69	97.209

**Table 15 sensors-23-08477-t015:** Test recognition rates of SVM for MNIST.

Training Size	5000	10,000	20,000	60,000
Two-stage	99.9074	99.9024	99.9376	99.9815
One-stage	97.8223	97.9693	98.1475	96.6545

**Table 16 sensors-23-08477-t016:** Accuracies with K = 8 and various sizes and epochs for MNIST.

Epoch/Training Size	5000	10,000	20,000
10	99.3978	99.5357	99.6363
15	99.1285	99.5958	99.7012
20	99.2428	99.5187	99.7647
25	99.2418	99.6123	99.7198
30	99.3296	99.4849	99.6489
50	99.0874	99.6048	99.6929
One-stage	97.8109	98.0697	98.1475

**Table 17 sensors-23-08477-t017:** Performance scores with one-stage and two-stage feature extractors.

	MNIST		USPS	
	SVM	MDC	SVM	MDC
One-stage	98.1475	97.1316	98.42	90.13
Two-stage	99.9815	97.2372	99.9863	91.60

**Table 18 sensors-23-08477-t018:** Comparison with state-of-the-art methods on MNIST.

Methods	Error Rates
LDANet-2 (Chan et al. [21])	0.62
PCANet-1 (L1′ = 64, k1′ = k2′ = 3) (Chan et al. [21])	0.62
Scatnet-2 (SVM rbf) (Bruno et al. [9])	0.43
Conv. Maxout and DropoutConv. Maxout and Dropout (Goodfellow et al. [23])	0.45
Stochastic pooling ConvNet (Zeiler et al. [24])	0.47
ConvNet (Jarrett et al. [8])	0.53
HSC (Yu et al. [26])	0.77
K-NN-IDM (Keysers et al. [27])	0.54
CDBN (Lee et al. [28])	0.82
KNN-SVM (Prasad et al. [39])	0.74
Deep Morph-CNN (Mellouli et al. [33])	0.34
NRC (Xu et al. [34])	1
**One-stage features on different classifiers**	
SVM	1.8525
MDC	2.8684
**Two-stage features on different classifiers**	
SVM	0.0185
MDC	2.7628

**Table 19 sensors-23-08477-t019:** Comparison with state-of-the-art methods of USPS.

Methods	Error Rates
NRC (Xu et al. [34])	4.90
Scatnet-2 (SVM rbf) (Bruno et al. [9])	2.30
IDM (Keysers et al. [27])	1.90
Online SVM learning (Tax et al., 2003 [44])	4.25
Discriminant-based supervised learning (Mairal et al. [30])	2.40
SVM KNN (Zhang et al. [31])	2.59
MQDF (Su et al. [32])	2.19
**One-stage features on different classifiers**	
SVM	1.58
MDC	9.87
**Two-stage features on different classifiers**	
SVM	0.0137
MDC	8.40

## Data Availability

Not applicable.

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
