# Peer review of "Two-Stage Feature Generator for Handwritten Digit Classification"

_sensors, 2023, doi:10.3390/s23208477_

Round 1
Reviewer 1 Report
In this paper, a new method of handwritten digits classification. The method is based on a novel feature generator framework which allows to recognise distinctive characteristics of digits. The paper is interesting but it has several limitations. The generator was proved for a small set signs, only digits. The digits have characteristics which make easier its recognition. Of course it doesn't lessen a meaning of the proposed method.
My remarks:
1. The paper needs a lot of editorial corrections which makes easier a reading of the paper. Especially it applies to equations, e.g. equation 1: what i and j mean? I assume that i and j are indexes but this is not visible from the notation.
2. Figure 2 should present more details of two stage feature framework generator. Authors presented very general their idea.
3. Page 5: How the MSE was calculated which results are presented on figure 3?
4. Equation 3 is incomprehensible.
5. An algorithm 4, second line: an abbreviation "mth" has to explained.
6. The same algorithm (case 1 and case 2): notation which applies to "sigma" has to corrected. Now the notation is incomprehensible.
7. References: Maybe item 40 is not mentioned on the text.
Main remarks:
1. Authors have to present shapes of analysed digits. Potential reader does not know anything what digits were analysed.
2. In my opinion the paper has to include tests results for other signs (not only digits) for which a similarity achieves higher values. I think that this testing area can’t be future works. These examinations should be conducted during this stage of works.
3. Conclusions - please to develop the sentence "In future work ...".
Author Response
We would like to thank the reviewer for his/her kind contribution to our study.

Reviewer 2 Report
This paper presents a Handwritten Numeral Recognition (HNR) system through features generated using Principal Component Analysis (PCA) and Partially Trained Neural Network (PTNN) and classification using Minimum Distance Classifier (MDC) and Support Vector Machine (SVM). The major observations are as follows:
1. Section 2: State of the Art is not organized and difficult to follow a long single paragraph. It is necessary to review related works especially those considered PCA and/or PTNN for HNR.
2. The last para of Section 2 does not match with the literature review may be transferred to the introduction. The last para after reviewing existing works might be observations from those.
3. Introduction section is not informative; a para having summary and contribution of the present study is necessary.
4. Major weakness of the paper is its presentation of the method. The distinction between traditional MLP/NN and PTNN is not clear. MDC is not popular but its description is not available. The methods should be explained straight forward way: feature generation and classification in different sections.
5. Paper holds wrong statements. In Page 4, ‘the PTNN module in the framework is simply a Multilayer Perceptron (MLP) [2] with a single layer with various neurons. It is structured for the purpose of classification.’ How single layer defines MLP? If it is structured for classification, why you need MDC and SVM?
6. In Page 4, the authors stated ‘Figure 3 illustrates Mean Squared Error (MSE) results obtained from the fully trained NN and PTNN training.’ But in the figure nothing is mentioned for NN and PTNN.
7. According to the heading of Algorithm 2 it is a feature generation algorithm then how classification using SVM. MDC in S6.
8. Again, the authors stated in Page 9 regarding MNIST is ‘the digits have sizes of 20x20 pixels.’ It is well-known that the MNIST digit image size is 28x28 pixels.
9. Presentation of the paper is poor. In general, a paragraph conveys a message and the first sentence is the topic sentence in a research paper. But many paragraphs in different pages are with single/two sentences and missed the norm. Typos are also available throughout the paper.
10. There are also several issues in the experimental result section. I think when a paper is not acceptable/justified from methodological points of view it is unnecessary to review its experimental outcomes.
The presentation is poor. English should be improved.
Author Response

(The authors gave the same response as above.)

Reviewer 3 Report
I find the purpose of the study to be significant and the quality of writing to be good. However, I have several major concerns that require addressing.
1. The contributions and motivations of the paper are not clear.
2.The authors should provide additional information to clarify symbols involved in this study.
3. More graphical results are suggested to demonstrated your findings.
4. The introduction is not structured in a coherent manner. It jumps between different topics without clear transitions, making it difficult to follow the argument.
5. The conclusions need to be improved and enlarged.
Author Response

(The authors gave the same response as above.)

Reviewer 4 Report
A two-stage cascaded feature generator framework is proposed for handwritten digit classification with PCA in the first stage and PTNN in the second. The features produced by the second one have been used to train MDC and SVM classifiers. Different experiments realized on MNIST and USPS show that the framework outperforms the state-of-the-art techniques
· Remarks and questions
o The introduction is incomplete; it poses the problem but does not indicate what your choice is in relation to the literature and the assumptions you make
o The state of the art is a narrative that tells the existing systems without any hindsight or synthesis. So, we do not see its interest and what are the lessons that we can draw from it
o What is the input in Figure 1?
o P6, l251: what does it mean: “where, B indicates between the classes”?
Author Response

(The authors gave the same response as above.)

Round 2
Reviewer 1 Report
Authors done a huge and a good work. Now a reading of the paper is easier. Authors explained my remarks but still the paper includes mistakes:
- Equation 4 has to be corrected (i and j indexes);
- A sharpness of Figure 3 should be higher;
- Table 2-5 and Tables 8-9, maybe comas are used incorrectly.
Author Response
We thank the reviewer for his/her kind support in improving the quality of the study.

Reviewer 2 Report
Authors updated paper but still it holds several major issues. Major observations are as follows:
1. The study is on Handwritten Numeral Recognition (HNR) but authors also reviewed a few character recognition studies (e.g., Lines 84-88 in Section 2). What is the motivation such review?
2. Presentation of the proposed method is not straight forward. Two stage feature generation (TSFG) is the main issue of the study, then Section 3.1One-Stage Feature Generator (OSFG) is misleading. The proposed method with two stage generation will be explain clearly. In experimental cases OSFG might be implemented to justify TSFG.
3. Experimental outcomes in Fig. 3 in proposed model description is unusual. Legend of Fig. 3 is not appropriate. Legend says two graphs but it is marking of partial training in full training. Again legend of the figure is not clear and epochs upto 2000 epochs (as mentioned in text) is not visible.
4. I mentioned in first revision, there are a lot for problem in presentation. Still numerous problems is available and not acceptable for any journal publication. Some points are
a. Presentation is not consistent: Section 2. State of Art but in introduction state-of-the-art. Author says MNIST and USPS but result presents for USPS first in several cases.
b. In Table 2, how ‘ 0,7689, 0,5903, 0,5389 and may values presented with comma (,) presents standard deviation. Similar issues are available in other problems.
c. Table 11 legend says Test recognition rate but in table no heading of it.
5. Paper is not written as well as checked carefully and it is difficult to mention all issues in review report.
Paper is not written as well as checked carefully, and it is difficult to mention all the issues in the review report.
Author Response
We thank the reviewer for his/her kind support in helping us to improve the quality of the study through his/her conservative comments.

Reviewer 3 Report
It is well revised. It can be accepted now.
Please check spelling, grammar, and spacing between words, before submitting final files.
Author Response
We thank the reviewer for his/her kind support in helping us to improve the quality of the study through his/her constructive comments.
